# Disturbance and climatic effects on red spruce community dynamics at its southern continuous range margin

Relena Rose Ribbons

Harvard Forest, Harvard University, MA, United States
Environmental Conservation Department, University of Massachusetts-Amherst, MA, United States
Bangor University, School of Environment, Natural Resources and Geography, Bangor, United Kingdom

Corresponding author
Relena Rose Ribbons,
rribbons@gmail.com

## ABSTRACT

Red spruce (*Picea rubens*) populations experienced a synchronous rangewide decline in growth and vigor starting in the 1960s, likely caused by climate change and a combination of environmental disturbances. However, it is not yet known if populations continue to decline or have recovered. Red spruce growing near its southern range margin in Massachusetts is a species of concern, in light of the vulnerability to climate change. This study uses population data from 17 permanent plots coupled with tree-ring data to examine radial growth rates, determine the growth-climate relationship, and document disturbance events. Red spruce at these plots ranged from 90 to 184 years old, and comprised 15 to 29 $m^2$/ha basal area. Red spruce seedlings and saplings were common at plots with previously high overstory spruce abundance, indicating it could return to a more dominant position under favorable growing conditions. However, permanent plot measures over a 50 year time span did not indicate any consistent trends for changes in basal area or density for red spruce or other woody species. Climate data show that mean annual minimum, maximum, and summer temperatures have increased over the last 100 years. Dendroclimatological analyses indicated that red spruce growth was sensitive to both temperature and precipitation. Prior to the 1960s, spruce at these sites showed a positive response to precipitation; however after a multi-year drought in the 1960s showed an increasingly negative correlation with precipitation. There has been a negative growth response to regional warming, as spruce radial growth was mostly constrained by increasing temperatures, potentially coupled with the associated increasing drought-dress. I suggest the change in climate response is potentially due to a physiological threshold response to increasing temperatures, which may cause spruce to continue to decline or be lost from the lower elevation sites, while the high elevation sites has a persistent spruce population.

## INTRODUCTION

Climate and global change are altering forests across the northeastern United States (*Loarie et al., 2009*; *Treyger & Nowak, 2011*). Climate change can profoundly alter forest ecosystems, including migration of species to maintain viable populations within specific-climate envelopes (*Petit, Hu & Dick, 2008*), and via altered seed production and regeneration rates due to increased carbon dioxide (*Mohan, Clark & Schlesinger, 2004*). Tree species at their southern range margins are likely to experience heat stress and exhibit reduced growth, with boreal species at southern limits most at risk because they are already living at the edge of their environmental tolerance. Evidence of vegetation communities changing with a warming climate in the eastern U.S. is already being observed (*Beckage et al., 2008*; *Mcmahon, Parker & Miller, 2010*; *Treyger & Nowak, 2011*). Decreased habitat suitability at southern latitudes, resulting from direct and indirect anthropogenic influences (e.g., non-sustainable forestry practices and climate change), might cause species to migrate north as climate becomes less favorable for their ideal growing conditions (*Allen, Macalady & Chenchouni, 2010*). In Massachusetts, red spruce forests are highly vulnerable to future climate change (*The Manomet Center for Conservation Sciences, 2010*). In New England, annual temperatures have risen an average of $+0.08 \pm 0.01°C$ per decade over the last century, and $+0.25 \pm 0.01°C$ per decade for the past three decades (*Hayhoe et al., 2007*). The greatest changes over the last 35 years have been in winter, with almost a degree C per decade increase (*Hayhoe et al., 2007*).

In the 1980s, red spruce (*Picea rubens* Sarg.) experienced widespread decline in vigor (*Hornbeck & Smith, 1985*; *Reams & Van Deusen, 1993*; *Scott et al., 1984*) and health throughout New England (*Battles et al., 2003*; *Siccama, Bliss & Vogelmann, 1982*; *Webb et al., 1993*), and in disjunct populations in the Southern Appalachians (*Cogbill & White, 1991*; *Silver et al., 1991*). Much of the research on red spruce within the Northeast focused on winter injury and the long-term influences of consecutive winter injury years (*Lazarus et al., 2006*) or the decline of high-elevation spruce-fir forests (*Battles et al., 2003*). Acid deposition effects on red spruce are well-documented (*Likens, Driscoll & Buso, 1996*), especially in high elevation forests where it is linked directly to red spruce mortality and indirectly affects red spruce via increased susceptibility to winter injury (*Lazarus et al., 2006*).

*Moore, Van Miegroet & Nicholas (2008)* found evidence of a rebound (increased growth or vigor) of high-elevation red spruce after declining in the southern Appalachians. Dendrochronology studies documented a sustained gradual decrease in radial growth measurements throughout red spruce's northern range (*Cook, Johnson & Blasing, 1987*; *Johnson, Cook & Siccama, 1988*; *Siccama, Bliss & Vogelmann, 1982*). Previous studies document decreased red spruce growth associated with high temperatures (*Cook & Johnson, 1989*). *Hamburg & Cogbill (1988)* noted red spruce displacement by hardwood species in the southern Appalachians.

Two potential hypotheses for the spruce decline emerged: (1) stand history factors such as pests, disturbance events including acid deposition, and natural stand aging processes; and (2) climate change factors such as increased temperatures and drought-stress. Multiple factors, such as climate and disturbance events, have contributed towards the widespread

decline of red spruce observed in many high-elevation populations (*Van Deusen, Reams & Cook, 1991*). While this decline seemed fairly ubiquitous in red spruce populations, it is unclear if these populations are still declining or have recovered from since the 1960s (*Gavin, Beckage & Osborne, 2008*; *Lazarus et al., 2006*), especially at low-elevations.

This study used red spruce tree-ring data coupled with population dynamics data documented in continuous forest inventory (CFI) plots, to explore the largely un-studied long-term dynamics of low-elevation (500–700 m a.s.l.) red spruce in western Massachusetts. I used tree-rings to investigate climate sensitivity to temperature and precipitation, the impact of climate change on spruce growth rates at their southern continuous range margin, and past disturbance history to elucidate the role of disturbance events and climate change as the mechanism behind observed changes in spruce growth. Based on prior studies indicating spruce's sensitivity to warm temperatures (negative correlations between temperature and ring-widths; *Cook, Johnson & Blasing, 1987*; *Johnson, Cook & Siccama, 1988*) and the recent warming documented in New England, I tested two hypotheses: (1) red spruce has declined in basal area and growth rates over the last 50 years in Massachusetts; and (2) red spruce growth rates were negatively correlated with mean maximum summer temperatures (heat stress).

## MATERIALS AND METHODS

### Study area

Red spruce in Massachusetts is located primarily in the western portion of the state (*Burns & Honkala, 1990*). Study sites were located in the Western Massachusetts Taconic Mountains and Berkshire Plateau region, including portions of four State Forests: Mount Greylock (MG), Savoy Mountain (SM), October Mountain (OM), and Middlefield Forest (MF) (Fig. 1). These four spruce-dominated forests were selected because they contained Continuous Forest Inventory (CFI) plots that were established in the 1960s to monitor the status of Massachusetts state forests over time (see below).

The climate is characterized by cold winters (average −5°C) and moderately warm summers (average 19°C). Average annual rainfall is 109 cm, 55% of which falls from April to September (*Scanu, 1988*). Average seasonal snowfall is 180 cm. The Berkshire region is underlain by igneous and metamorphic rocks, especially gneiss and schist (*Scanu, 1988*). Mount Greylock is comprised mainly of granite and limestone bedrock. Soils are primarily rocky glacial tills, with sand and gravel deposits proximate to waterways (*Motts & O'Brien, 1981*). Plots at Mount Greylock are located at higher elevations (678–978 m a.s.l.) than the three other forests (540–724 m a.s.l.; Table 1). Field sites were established with permission from the Department of Conservation and Recreation undertaken with a special use permit.

### Site selection and vegetation characterization

I selected seventeen field plots from a subset of 0.08 ha circular CFI plots established every 0.8 km within a square grid across all State Forest Lands in 1960, except site MF which was established in 2000. Since only one plot exists within MF (*n* = 1), MF was excluded

**Peer**J

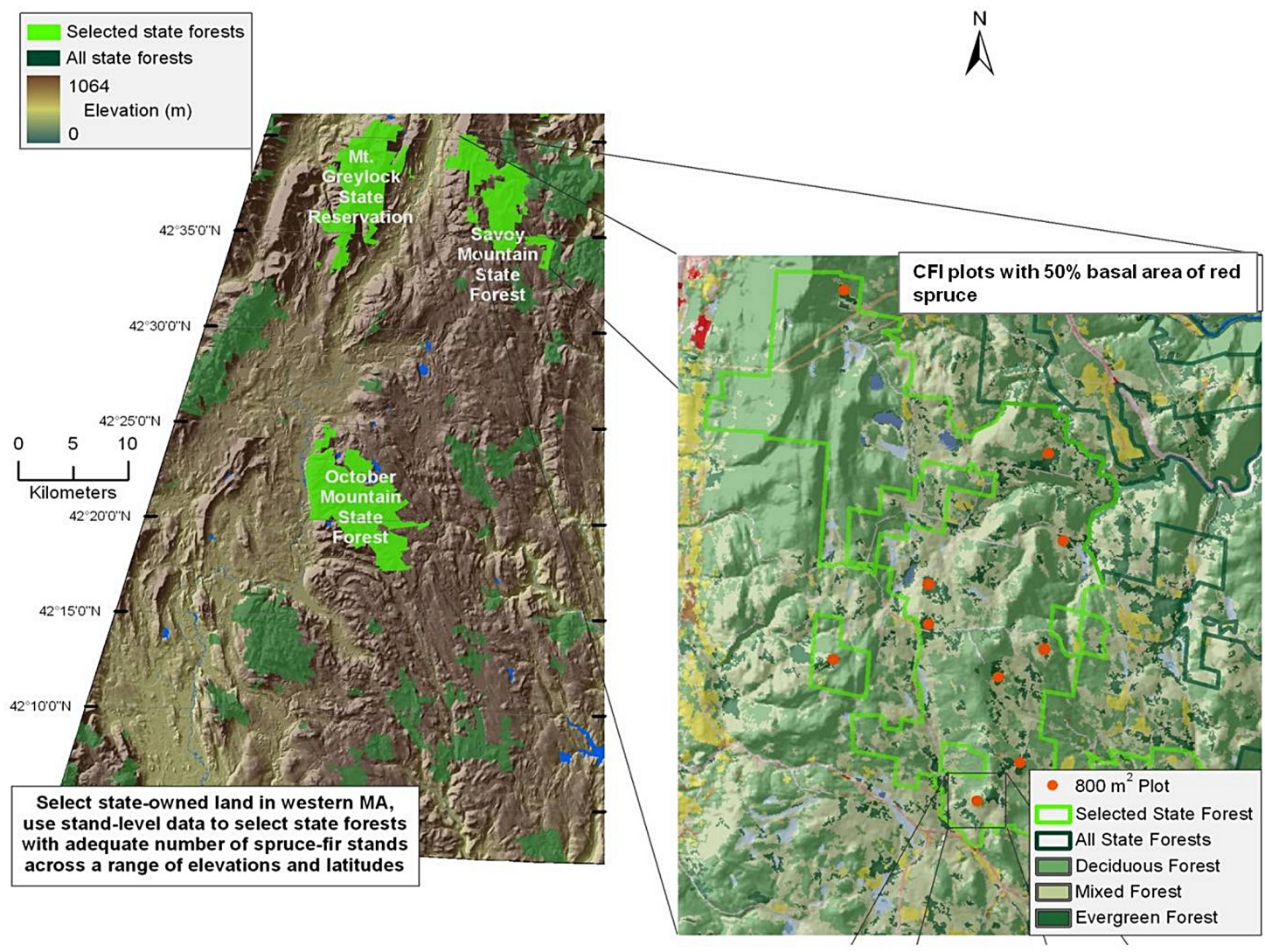

**Figure 1  Map of sampled forests and red spruce distribution.** Map of site locations in western Massachusetts, with key forests highlighted in green. Inset shows the complete range of red spruce.

from all stand dynamics statistics. Plots located on upland sites and containing forests characterized by at least 50% spruce-fir or red spruce basal area at the time of the last forest survey in 2000, were used for this study. The 17 plots were relocated using a handheld GPS unit and plot boundaries were re-established for the vegetation inventory from May through September 2010. At each CFI plot, all overstory trees >12.7 cm diameter at breast height (DBH; 1.37 m) were measured and identified to species. Efforts were made to relocate previously numbered or painted overstory trees within each plot, to match exact tree measurements with past inventories; however, paint had worn off most trees within plots so individual tree diameter comparisons were not made. All saplings (5–12.7 cm DBH) within the plot were tallied and identified to species. In addition, percent cover of

Table 1 **Location and plot information for each CFI forest.** Location (decimal degrees) of each CFI forest. Note the number of plots at each forest.

|  | Mount greylock | October mountain | Savoy forest | Middlefield forest |
|---|---|---|---|---|
| Latitude | 42.622364 | 42.358235 | 42.594551 | 42.401972 |
| Longitude | −73.181151 | −73.1771771 | −73.0288963 | 73.0808754 |
| Elevation | 678–959 m | 540–613 m | 528–724 m | 542 m |
| Number of plots | 7 | 5 | 4 | 1 |

seedling and shrub species were estimated in a 5 m radius subplot located in the plot center, to determine regeneration of red spruce and other tree species. Dead spruce trees were tallied, although individual tree-mortality comparisons were not made due to the loss of individual tree markers and painted numbers. Total basal area, red spruce basal area, and red spruce density were calculated for each sampling period (1960, 1980, 2000, and 2010) using past CFI data. Additionally, basal area and density of all trees except red spruce were calculated for each sampling period. Relative importance values were calculated at each site as the sum of relative basal area and relative density for each species. Stand dynamics data were analyzed using linear mixed models in *R Core Team (2013)* with the nlme package (*Pinheiro et al., 2013*). Nomenclature follows *Gleason & Cronquist (1991)*.

## DENDROCHRONOLOGICAL METHODS

### Field data collection

All overstory red spruce (>10 cm DBH) trees within each plot were cored at DBH with an increment borer. When <10 red spruce occurred within a plot, additional red spruce located outside the plot were cored until at least 10 trees were obtained from each site). Two cores per tree were extracted to identify any locally absent rings and to provide a better estimation of whole-tree radial growth. 213 cores were successfully cross-dated and used for disturbance and climate analysis.

### Sample preparation and measurements

All cores were dried, mounted, and sanded using progressively finer sandpaper, up to 600-grit. Each core was visually crossdated within each tree first using the list method (*Yamaguchi, 1991*) and cores were compared among trees within each plot. Cores were then measured to 0.001 mm precision using to an Olympus SZ40 microscope and a Velmex measuring system (Velmex, East Bloomfield, NY, USA) interfaced with Measure J2X software. Ring widths were used to statistically verify the accurate dating of all tree rings using Spearman's correlation analyses (with a critical $r$-value of 0.33) available in the COFECHA tree-ring software (*Holmes, 1981*).

### Data analysis

After verification using COFECHA, red spruce chronologies were constructed for each forest using the program ARSTAN (*Cook, 1985*). A window of 32 years, with 16-year overlaps was used for the common period of 1930–2009, with a variance stabilization

of one-third the length of the longest core. Data were first detrended with a negative exponential curve, then detrended with a −2/3 smoothing spline, with interactive detrending used on individual cores when necessary (e.g., when growth trends appeared inflated by poorly fitted detrending curves). Residual chronologies from ARSTAN-averaged ring-width indices across an entire sampling site with autocorrelation removed-was used for climate analysis.

To track the climate response over time, residual ARSTAN chronologies were correlated against mean monthly maximum and minimum temperature and precipitation from the prior March to October of the current year from 1896 to 2009. Climate data from the PRISM project (*Prism Climate Group, 2011*) were correlated with tree ring widths using Dendroclim2002 moving interval analysis to analyze the chronologies from the previous March to the current October months using a moving window of 40 years (*Biondi & Waikul, 2004*). Only significant correlation values are plotted in the DendroClim2002 output, which has red to indicated significantly positive values, and blue to indicate significantly negative values. To assess whether sites exhibit temporal stability in their climate response I examined the DendroClim output graphs looking for consistent colored bars over the entire time period which suggests a temporally stable response, or vice versa to establish a temporally non-stable climate response. I also used response function analysis (RFA), a multiple regression technique that uses principal components of monthly climate data to estimate indexed values of ring widths and examine how climate influenced radial growth over the entire time period (*Fritts, 1976*). RFA was implemented in DendroClim and uses bootstrapping techniques to determine significance levels ($\alpha = 0.05l$; *Speer, 2010*).

Ring width chronologies were used to calculate average yearly growth rates in basal area increment (BAI expressed as mm$^2$/year). To calculate BAI, the radius of an individual tree is multiplied by the number Pi (3.14) to equal the area of the circle. The annual increment for year $X$ is defined by the following equation: $X − (X − 1)/(X − 1)$, where $X$ is the basal area at year $X$ (last year of growth) and $X − 1$ is the basal area of the tree measured up to the year previous to $X$ (*Johnson & Abrams, 2009*).

Stand disturbance dynamics within each forest were also reconstructed by evaluating all cores separately for abrupt growth releases using the criteria established by *Lorimer & Frelich (1989)*, who defined a "major sustained release" as an average growth increase ≥100% lasting at least 15 yr relative to the previous 15 yr and a "moderate temporary release" as an average growth increase ≥50% lasting 10–15 yr relative to the previous 10–15 yr. In addition, abrupt growth decreases ≥50% lasting 10 yr relative to the previous 10 yr were tallied for each canopy tree as an indication of canopy damage (*D'Amato & Orwig, 2008*). The use of 10- and 15-year windows within these release criteria removed the effects of short-term growth responses due to climatic events such as droughts (*Lorimer & Frelich, 1989*; *Nowacki & Abrams, 1997*). Disturbance chronologies were constructed for each study area by tallying the number of release and damage events by decade (*Lorimer & Frelich, 1989*). Disturbances were only counted from one core per tree (*Copenheaver et al., 2009*).

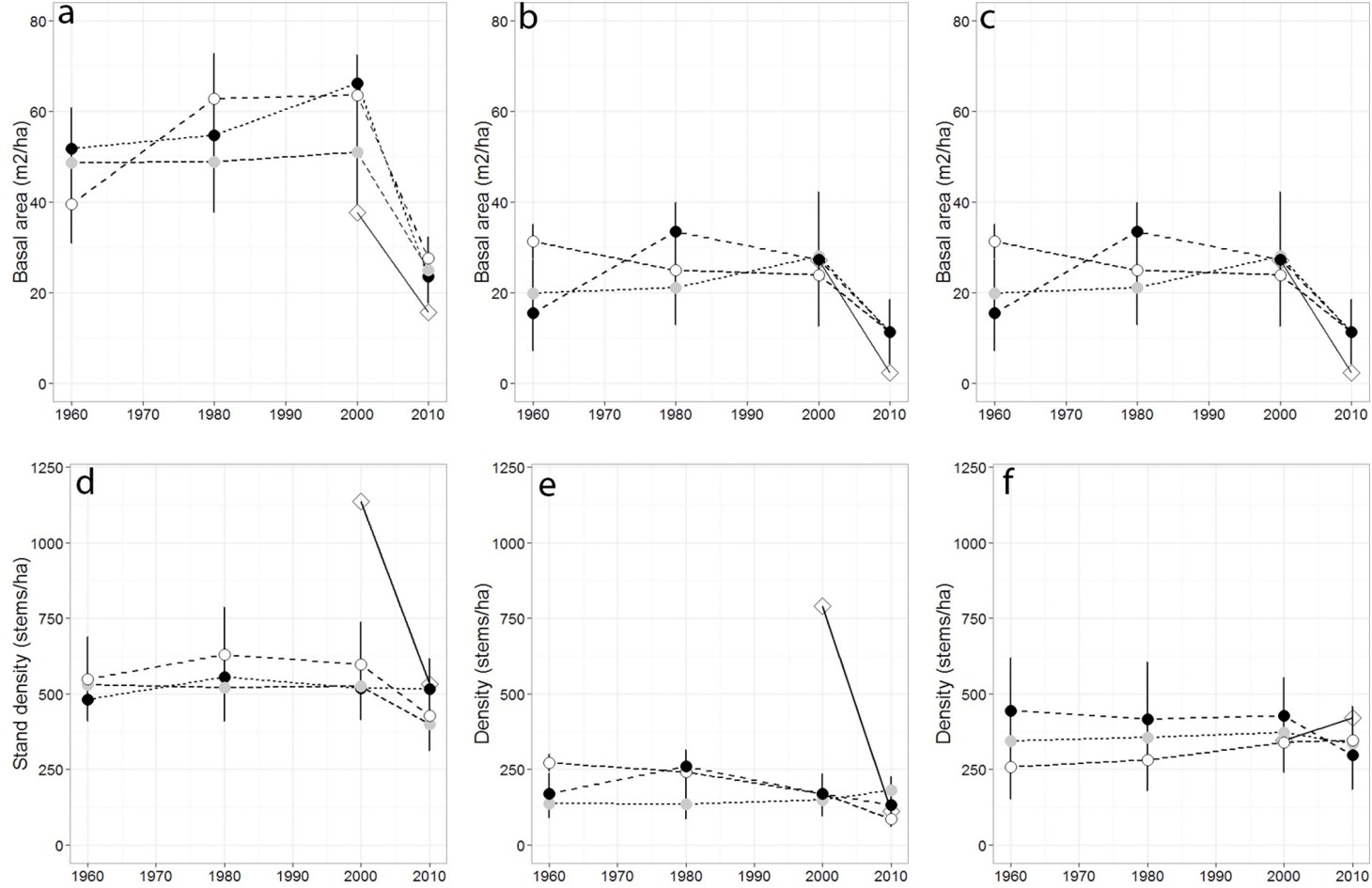

**Figure 2 Basal area and density for all trees, red spruce only, and non-red spruce trees among the four forests over time.** (A) Total basal area of trees ($m^2$/ha), (B) basal area of red spruce, (C) basal area of non-spruce species (D) Total stand density (stems/ha), (E) density of red spruce, and (F) density of non-red spruce in MG (black circle), OM (open circle), SF (grey circle), and MF (open diamond) forests across sampling years.

## RESULTS AND DISCUSSION

### Stand composition

Over the past 50 years, total stand basal area differed across sampling years ($p = 0.04$) but not among forests ($p = 0.079$), while total stand density remained largely stable across sampling years ($p = 0.45$; Figs. 2A–2G), and among forests ($p = 0.63$). Red spruce basal area differed significantly over time ($p = 0.02$) but remained relatively constant at MG, increased at SF, and decreased at OM and MF forests, and did not significantly vary among forests over time ($p = 0.191$). Basal area of all other species did not significantly differ among forests ($p = 0.71$), sampling year (0.20), or interact over time ($p = 0.23$). Non-spruce basal area declined by at least 50% at MG and MF forests, increased slightly at SF and increased by over 100% at OM. Red spruce density differed significantly over time ($p = 0.02$) among forests ($p = 0.005$), with declines at OM and SF forests, while MG increased in density (Fig. 2, Table 3).

**Table 2 Stand dynamics for overstory and understory tree species across CFI forests.** Species overstory relative importance values (IV; relative density * relative dominance )/ha, and the most common sapling densities in the CFI forests.

| | MG | OM | SF | MF |
|---|---|---|---|---|
| | 2010 (%) | 2010 (%) | 2010 (%) | 2010 (%) |
| **Overstory species** | | | | |
| *Abies balsamea* | 11.8 | 6.0 | 1.7 | 6.6 |
| *Acer rubrum* | 8.5 | 19.8 | 26.4 | 33.2 |
| *Acer saccharum* | 3.7 | — | 6.9 | — |
| *Betula alleghaniensis* | 13.2 | 6.0 | 5.7 | 11.4 |
| *Betula papyrifera* | 8.4 | 4.6 | 2.5 | 4.4 |
| *Fagus grandifolia* | 5.9 | 2.2 | 8.2 | — |
| *Fraxinus americana* | — | 0.4 | 7.8 | — |
| *Picea rubens* | 39.8 | 39.2 | 30.7 | 15.2 |
| *Prunus serotina* | 4.4 | 3.4 | 3.6 | — |
| *Tsuga canadensis* | — | 10.3 | 5.9 | 29.3 |
| Other species | 2.2 | 8.0 | 0.6 | — |
| **Saplings (Density/ha)** | | | | |
| *Abies balsamea* | 185.4 | 154.5 | 617.9 | 12.4 |
| *Acer pensylvanicum* | 119.5 | 12.4 | 185.4 | — |
| *Acer rubrum* | — | 185.4 | — | 49.4 |
| *Betula alleghaniensis* | 49.4 | 129.8 | 80.3 | 37.1 |
| *Fagus grandifolia* | 61.8 | 271.9 | 278.0 | — |
| *Ilex verticillata* | — | — | — | 74.1 |
| *Picea rubens* | 262.0 | 143.4 | 216.3 | 74.1 |
| *Pinus strobus* | — | 185.4 | — | — |
| *Tsuga canadensis* | — | 86.5 | — | — |

Sharp declines in red spruce densities were observed at OM after 1980 (272 to 168 stems/ha) and at MF since the 1980s (790 to 111 stems/ha; Table 2). The density of all other species declined 20–50% at three of the four study sites between 2000 and 2010; however the density of other species did not significantly differ since 1960 either by forest ($p = 0.80$) or sampling year ($p = 0.72$). At MF forest, the density of species other than spruce increased approximately 15%. Paper birch relative importance increased in each forest, while American beech and red maple increased in importance at three of the forests (Table 2). Saplings present within sampled forests were largely American beech, red maple, and balsam fir. Some sites still supported a healthy population of understory red spruce, indicated by abundant red spruce saplings (216–262/ha).

### Radial growth and climate response

Mean maximum, minimum, and mean summer temperatures have increased over the past century at all sites. There has been an average increase of 1°C over the past century in mean temperature (Fig. S7). Precipitation remained relatively stable with a slight increase over the past century, except for a prolonged decrease in total precipitation during a lengthy

**Table 3 Stand basal area and density among forests over time.** Mean (± standard error) stand, red spruce, and non-spruce basal area and density within MG, OM, and SF forests over four sampling years. MF was excluded since it has 1 CFI plot established in 2000.

| Forest | Year | Total.Stems | Spruce.Stems | Non.spruce.Stems | Total.BA | Spruce.BA | Non.spruce.BA |
|--------|------|-------------|--------------|------------------|----------|-----------|---------------|
| MG | 1960 | 87.63 (± 15.86) | 137.99 (± 49.37) | 76.43 (± 15.73) | 9.41 (± 1.92) | 19.85 (± 7.42) | 7.09 (± 1.44) |
| MG | 1980 | 86.5 (± 15.80) | 133.87 (± 48.75) | 76.35 (± 15.95) | 9.64 (± 2.01) | 21.11 (± 8.24) | 7.19 (± 1.42) |
| MG | 2000 | 86.5 (± 18.65) | 148.29 (± 52.62) | 74.15 (± 19.41) | 11.03 (± 2.56) | 28.06 (± 11.10) | 7.63 (± 1.69) |
| MG | 2010 | 88.27 (± 21.98) | 181.24 (± 44.06) | 69.03 (± 23.65) | 4.03 (± 0.99) | 11.26 (± 4.11) | 2.54 (± 0.59) |
| OM | 1960 | 109.51 (± 20.72) | 271.87 (± 27.26) | 67.16 (± 15.84) | 10.06 (± 2.31) | 31.22 (± 3.87) | 4.54 (± 0.97) |
| OM | 1980 | 98.86 (± 21.15) | 225.97 (± 62.88) | 64.64 (± 15.83) | 9.85 (± 1.98) | 25.98 (± 4.39) | 5.51 (± 1.26) |
| OM | 2000 | 89.12 (± 19.46) | 175.48 (± 61.84) | 73.70 (± 19.23) | 8.21 (± 2.20) | 22.21 (± 10.16) | 5.71 (± 1.58) |
| OM | 2010 | 72.09 (± 11.66) | 86.5 (± 27.07) | 69.20 (± 13.03) | 4.94 (± 1.10) | 11.32 (± 3.12) | 3.67 (± 1.04) |
| SF | 1960 | 68.48 (± 14.78) | 168.89 (± 69.30) | 54.14 (± 11.68) | 4.93 (± 1.38) | 15.53 (± 8.40) | 3.41 (± 0.75) |
| SF | 1980 | 75.63 (± 17.26) | 259.51 (± 12.36) | 50.55 (± 11.64) | 7.52 (± 2.19) | 33.41 (± 6.41) | 3.99 (± 0.85) |
| SF | 2000 | 74.66 (± 15.52) | 168.89 (± 66.29) | 61.20 (± 13.47) | 7.94 (± 2.39) | 27.31 (± 14.79) | 5.17 (± 1.15) |
| SF | 2010 | 67.64 (± 12.37) | 131.81 (± 53.55) | 55.61 (± 9.23) | 4.35 (± 1.28) | 11.28 (± 7.08) | 3.05 (± 0.61) |

drought in the 1960s (Fig. S6). Raw red spruce ring widths for each forest (Fig. S1) show trends of declining radial growth rates over time, which is consistent with natural decreases in growth as trees mature. Detrending was used to remove growth-trends which extend beyond a 32 year frequency, so as to emphasize decadal-to-annual variability in tree growth rates. This process rules out natural declines in ring-widths due to age and suggests an external force (such as climate or environmental disturbance) limited spruce growth. The ARSTAN chronologies (Figs. S2 and S3) are more conservative in response to climate parameters, and were used for remaining analyses.

Basal area increment (BAI) was calculated using both cores from a tree, averaged across plots and within forests (Fig. 6). BAI curves show that on average, SF and OM forests declined over the last century, while MF and MG increased in BAI. Response Function Analysis (RFA) for regional climate data showed a significantly positive response of spruce radial growth with maximum monthly temperatures in May, and negative responses with maximum temperature in July and October precipitation (Fig. 6). RFA responses to temperature and precipitation differed across the four forests (Figs. S8–S11), with specific trends explained below.

## Forest-level trends in BAI and RFA

BAI at MG has continued to increase suggesting that red spruce at these sites may be locally adapted to stressful conditions and are more resilient to climatic changes. Red spruce are likely to continue to demonstrate positive growth at MG. In contrast, OM BAI declined for a decade from the mid-1970s to 1980s and again from 1998 to present, reinforcing the theory that red spruce at this forest are declining. SF BAI declined from the mid 1960s to mid 1980s, where it began to increase until 1998 when these spruce experienced a dramatic reduction in BAI. Since 1998 SF red spruce BAI has increased but it still quite variable. MF BAI has steadily increased, suggesting red spruce at this forest experience favorable growing

conditions. Taken together, only MG and MF forests have increasing BAI, while OM and SF show trends of declining BAI.

Similar to the regional response, MG RFA had a significantly positive relationship with maximum temperatures in May and a negative relationship with precipitation in October (Fig. 7). OM RFA showed spruce was negatively related to maximum temperatures in July and positively related with minimum temperatures in December, and had no significant growth response to precipitation (Fig. 7). SF RFA had a significant response to November maximum temperatures, with no significant responses to minimum temperatures and precipitation (Fig. 7). MF RFA had a significant positive response to precipitation in August, and no significant responses to temperatures (Fig. 7).

## Temporal stability in climate-growth response at a regional and forest-scale

Moving interval analysis (using a moving 40 year window) with the regional ARSTAN chronologies showed that spruce growth had a consistent positive correlation (as indicated by dark red bards) with wintertime temperatures from the late 1930s until the late 1960s. After the 1960s spruce growth was then negatively correlated with the previous years' summer temperatures until the early 1990s, after which it was positively associated with previous years' May temperatures (Fig. 5). Regional spruce growth was positively associated with current years' August precipitation for the last 3 decades. Red spruce growth at MG was positively correlated with average maximum and minimum winter temperatures from November of the previous year to February of the current growing year, from 1936 until 1966 (Fig. 3). After 1956, radial growth was negatively correlated with maximum March temperatures from the previous year, as indicated by the dark blue bars, remained consistently negative until 1986, after which the relationship was no longer significant. Previous year's July temperatures showed a marginally significant negative relationship with growth beginning in 1966 to the mid-1990s at MG (Fig. 3).

The OM ARSTAN chronology showed a positive relationship to minimum and maximum winter temperatures from the previous November to the current February (Fig. 3). I observed a strong negative response with spruce growth and current and previous summer minimum temperatures from 1970 to around 1990 and a similar response to previous year's maximum temperatures after 1986. The SF ARSTAN chronology showed a positive relationship between radial growth and minimum February temperature from 1960 until 1984 and with minimum September temperatures from the early 1970s until the late 1990s (Fig. 3). A strong negative correlation between growth and current year July maximum temperatures was observed until the mid-1980s after which the previous year's July maximum temperatures show a consistent negative response (Fig. 3). Previous March minimum temperatures were also negatively correlated with radial growth, which was consistent until the 1990s (Fig. 3), when the relationship deteriorated. MF exhibited weak correlations between growth and maximum temperatures, with no strong patterns emerging (Fig. 3). The MF residual chronology was positively correlated with both current and previous year's September minimum temperature from the 1930s until the mid-1960s, after which there was no significant relationship.

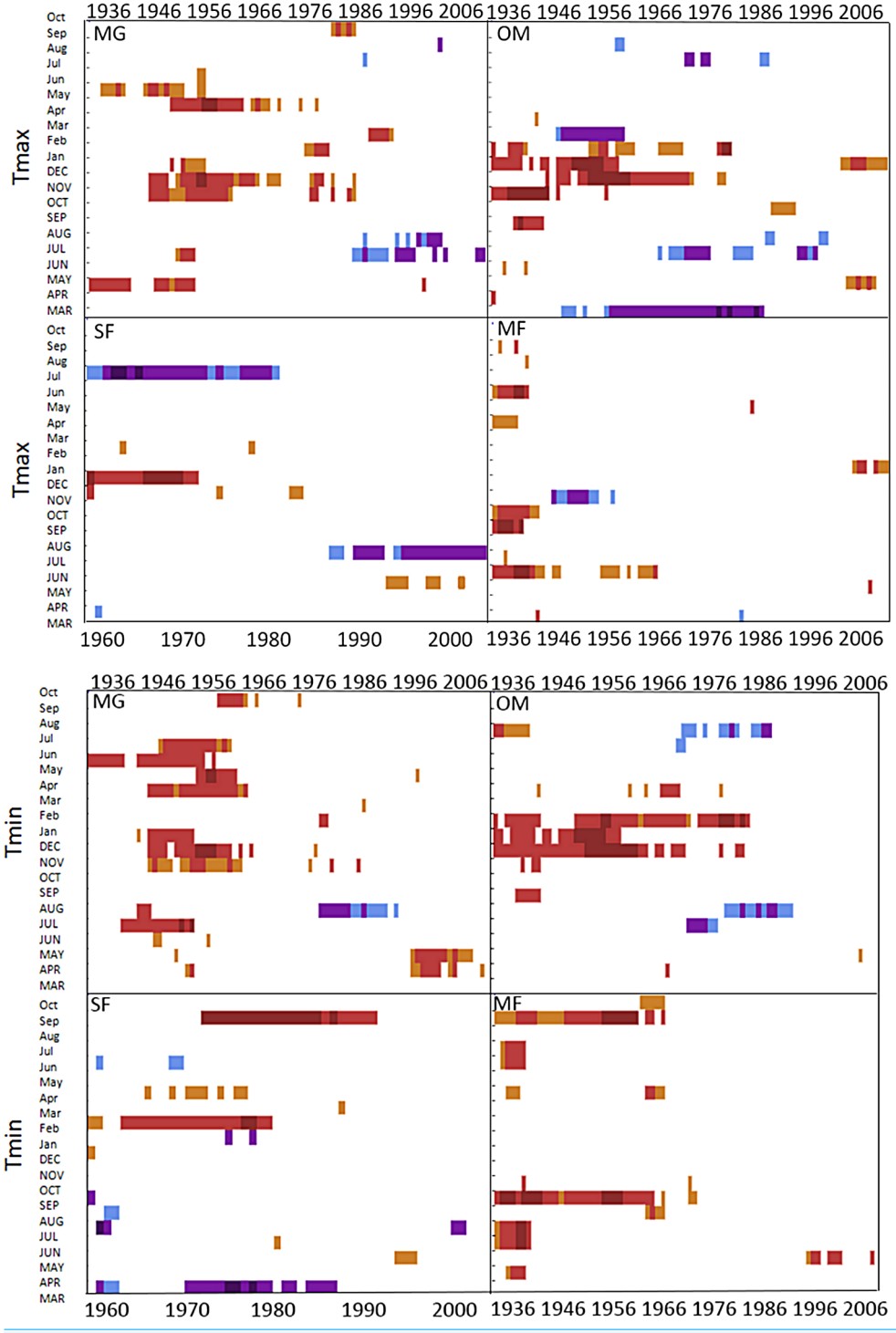

**Figure 3 DendroClim maximum and minimum temperature correlations with forest chronologies.** Moving interval analysis graphs from DendroClim showing correlations between a site's ARSTAN chronology and maximum (top four graphs) and minimum (bottom graphs) monthly temperatures for that site.

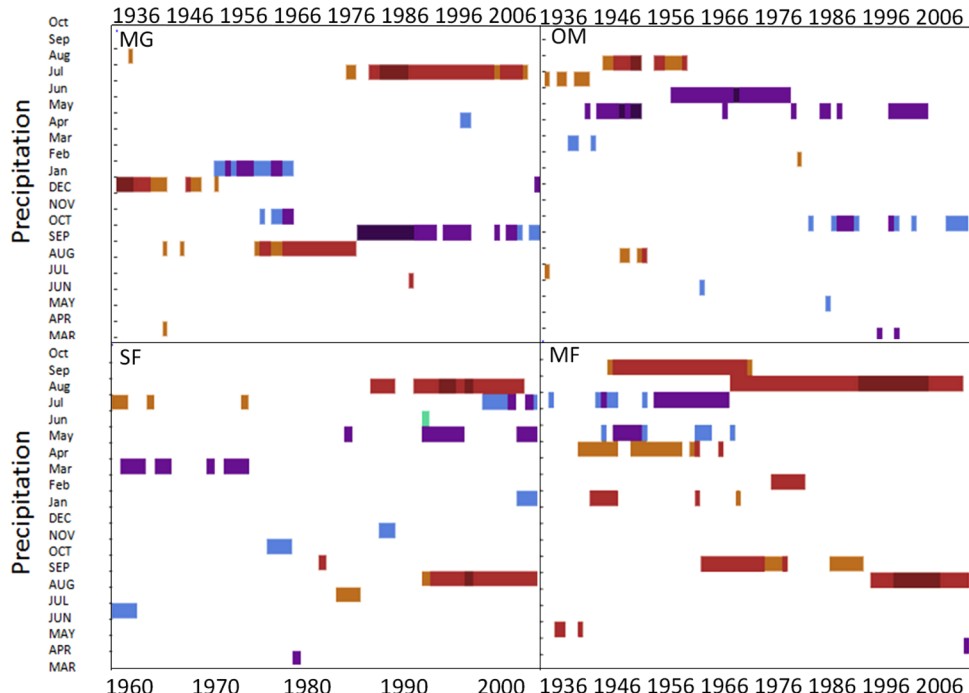

**Figure 4 Dendroclim precipitation correlations with forest chronologies.** Moving interval analysis graphs from DendroClim showing correlations between a site's residual chronology and monthly precipitation for each forest.

Spruce growth relationships with precipitation varied among sites. Until 1956, MG growth was positively correlated with January precipitation (Fig. 4) and then after 1976 was negatively correlated with previous year's October precipitation. From the mid 1960s and 1970s to the present, spruce growth was positively correlated with current August precipitation at the MG, MF, and SF sites and with previous August precipitation at both the MF and SF sites. No strong correlations existed between precipitation and radial growth at OM for more than a few years, except for a negative correlation with June precipitation from the mid-1950s to the late 1970s, suggesting that precipitation was not the primary limiting factor influencing radial growth at this site (Fig. 4).

## Disturbance history

The four sites I examined experienced varied radial growth dynamics across a gradient of disturbance (Fig. 8). MG experienced the most disturbance events with an average of 1.08 moderate, 0.58 major, and 1.30 declines per tree, followed by OM with 0.75 moderate, 0.63 major, and 1.65 declines per tree, SF with 0.67 moderate, 0.36 major, and 1.22 declines per tree, and MF with 0.35 moderate, no major, and 1.30 declines per tree. Prior to 1900 sampling depth was low, so I did not attempt to interpret detailed stand histories during this time. Following 1900, a high % of spruce experienced moderate growth releases in the 1920s and 1930s and both moderate and major growth releases in the 1970s and 1980s (Fig. 8). Low-level disturbance was detected in most other decades. Significant periods of growth decreases occurred at MG during the 1920s, 1960s, and the 1990s. At MG, OM, and

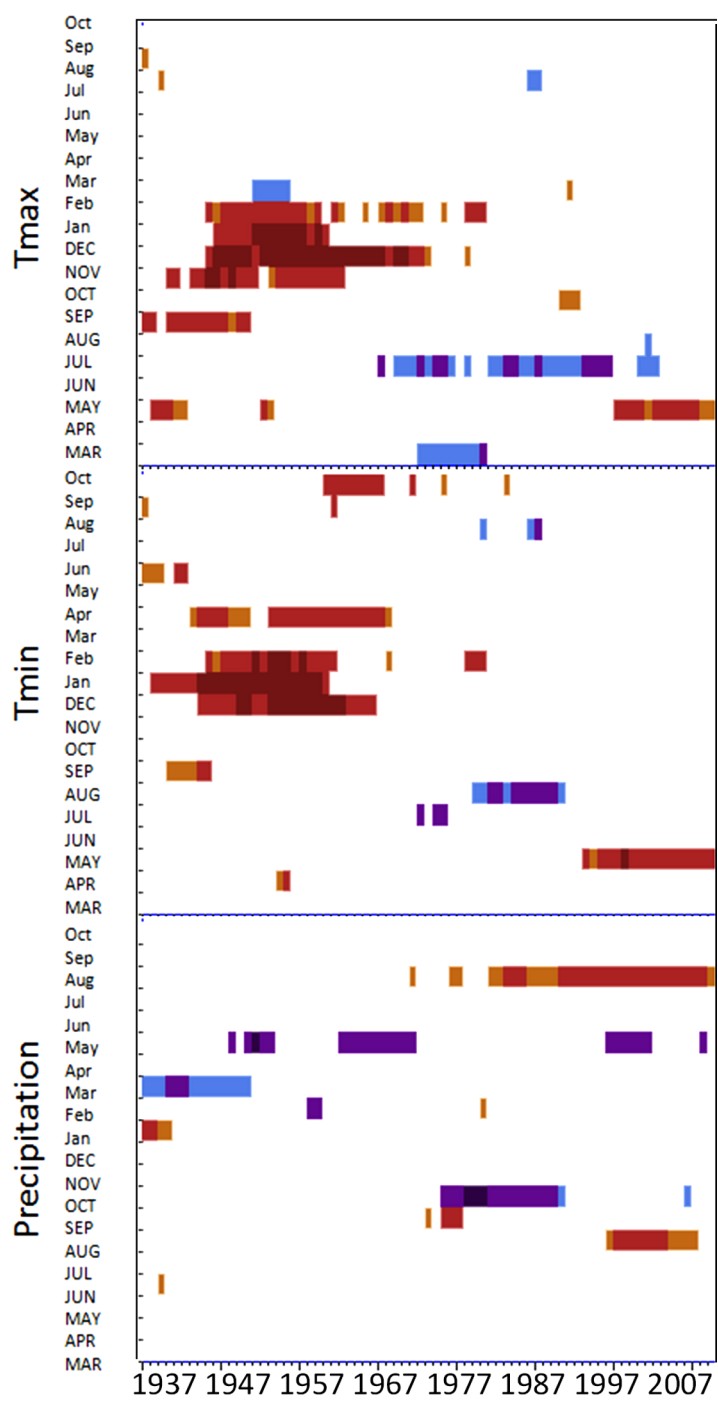

**Figure 5 DendroClim correlations with temperature and precipitation with the regional tree-ring chronology.** Regional ARSTAN moving interval analysis in DendroClim, showing correlations across all forests combined into one chronology correlated with the regional climate (maximum and minimum temperatures, and precipitation).

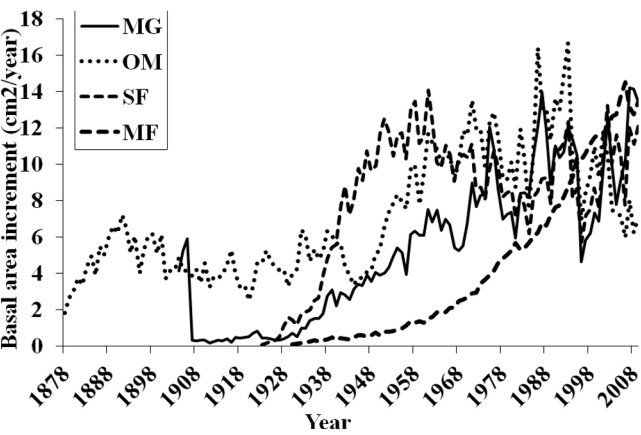

**Figure 6 Basal area increments for each forest.** Basal area increment averages for each forest with calculations truncated when fewer than two core replicates.

SF, I observed a pulse of moderate and minor disturbance events in the 1930s, which is the decade of the 1938 Hurricane (*Boose, Foster & Fluet, 1994*; *D'Amato, Orwig & Foster, 2008*). During this decade, disturbance events were noted in 25% of trees at MG, 22% of trees at OM, and 16% of trees at SF.

There was a pulse of major growth increases seen in OM spruce cores during the 1930s and 2 periods of moderate increases in the 1920s and 1980s. Significant growth reductions were seen in OM spruces during the 1950s, 1970s, and especially the 1990s, when all sampled trees exhibited growth reductions during this decade. This is consistent with the stand dynamics hypothesis for red spruce decline as supported by *Reams & Van Deusen (1993)*, which contends that the new growth released in the 1920s and 1930s at these sites naturally slowed in growth and increased competition led to some mortality. SF spruce experienced growth increases during the 1940s and 1950s and over 80% of trees experienced growth declines during the 1960s. Unlike the other sites, very little disturbance was detected in MF spruce cores, including no major growth releases in any decade examined. The most dramatic period of growth declines occurred at 3 of the 4 forests during the 1990s, when 70–100% of the trees experienced sharp growth reductions.

Over the last 50 years, some of the red spruce forests in western Massachusetts have declined in radial growth and density while maintaining basal area in the overstory. At some forests the understory is largely not red spruce seedlings and saplings, but instead a variety of northern hardwood species. Basal area increment has declined at some forests (OM and SF), but others (MG and MF) have steadily increased and do not exhibit extensive declines in basal area increment. Some red spruce trees have shifted their climate-growth responses, as demonstrated by moving interval analyses and the temporally unstable relationships between temperature, precipitation and ring widths. Red spruce in this study were located at lower elevations (500–700 m a.s.l.) than sites studied in the Adirondacks, Vermont, and Appalachian mountains. Additional studies in low-elevation red spruce populations will help to discern whether climate or physiology contributes to the red spruce growth decline widely observed in the 1960s–1980s.

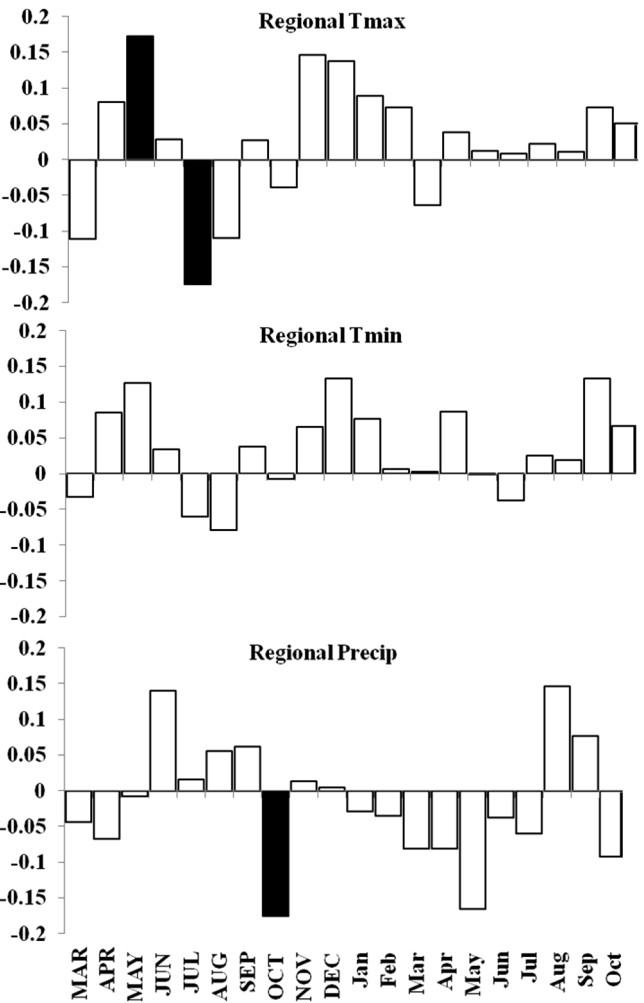

**Figure 7 Response function analyses for regional ARSTAN chronology with maximum and minimum monthly temperatures and precipitation.** Response function analyses for the regional ARSTAN chronology showing the period response for red spruce over previous year's climate parameters (months shown in capitals) and current year's climate parameters, with black bars indicating significant responses with maximum and minimum temperatures and precipitation (Tmax, Tmin, Precip).

Red spruce populations at some forests have sustained radial growth declines since the 1940s–1960s period (see raw-ring widths for each forest in Fig. S1). Tree growth rates naturally decrease as they age or increase in size (*Fritts, 1976*), so a trend of declining radial growth or ring increment is expected; however since these populations are not very old, the patterns of declining ring-widths are not explained by allometry and are not common to all populations. During the 1960s the Northeast experienced a severe drought, which could have contributed to spruce water limitation. Some individual trees rebounded in radial growth beginning in the 1990s (Figs. S2 and S3); likely due to stand disturbance events such as growth releases and decreased proximate competition within the stand.

MG, OM, and SF showed a common trend of sudden growth declines in the 1960s through the 1980s which corresponds the observed widespread red spruce decline in

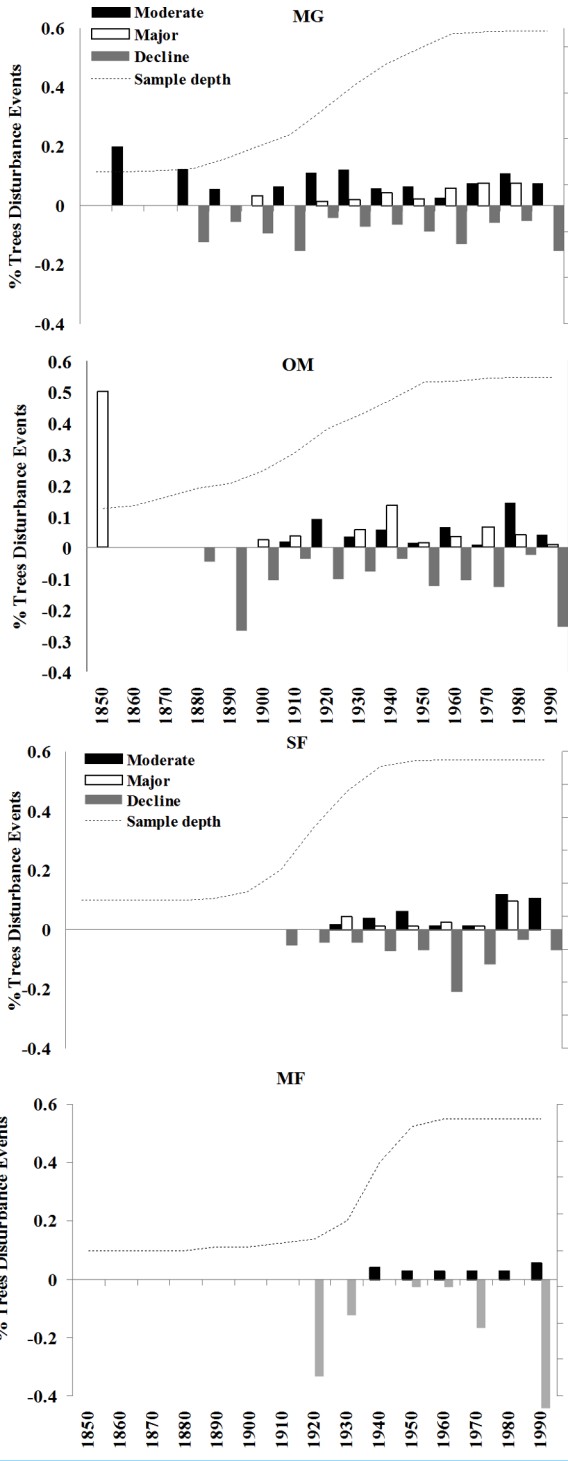

**Figure 8 Decadal disturbance chronologies for each forest.** Disturbance chronology with decadal distribution of radial growth increases and abrupt decreases (see Methods for criteria) measured in tree cores from four red spruce stands in western Massachusetts. Dashed line represents the sampling depth at each site.

the northeast. The highest maximum temperatures were observed during the 1990s and the residual ARSTAN chronology also shows the lowest values for spruce radial growth during this time period, supporting the theory that the increased temperatures have led to decreased radial growth. Despite mortality of older red spruce, MG sites show continued spruce establishment and recruitment, which is why stand density remained consistent, which is consistent with recent findings at other high-elevation or higher latitude sites (*Van Doorn et al., 2011*; *Kosiba et al., 2013*). In contrast, several OM sites have declined in both overstory and understory spruce density, reducing the likelihood of spruce retention. SF and MF sites have maintained a red spruce canopy but have little to no spruce regeneration, making it less likely that red spruce will persist.

### Non-stable climate growth relationships

Temperature had a much stronger influence on red spruce radial growth than precipitation over time across the region and the 4 forests (Figs. 3–5), although this relationship was not temporally stable. While a consistent climate-growth relationship was observed across the Massachusetts sites during the first half of the twentieth century, a distinct shift in this relationship began in the 1960s, but was variable across sites. Non-stable climate relationships can be explained by either climate change or a physiological tolerance/threshold response as suggested by *Cook & Johnson (1989)*. The physiological threshold response can be best explained as stress-induced physiological changes within trees that alter how they grow in response to climate (e.g., after an extreme drought trees may decrease their water uptake and efficiency; *D'Arrigo et al., 2008*; *Driscoll et al., 2005*; *Wilmking et al., 2005*; *Wilmking et al., 2004*). Studies have documented unstable climate responses for other red spruce populations (*Johnson, Cook & Siccama, 1988*; *Conkey, 1986*; *Conkey, 1988*) and several other boreal species (*Andreu-Hayles et al., 2011*; *Shi et al., 2010*) and attributed them to stand history events such as logging or past land use which were consistent across sites. The variability in responses across these sites supports and climatic-threshold response, since my disturbance history analysis shows periods of release but not consistently across sites.

Changes in maximum and minimum winter temperatures influenced red spruce radial growth over the past century at these sites. Prior to the 1960s, spruce growth was positively associated with minimum and maximum winter temperatures (Fig. 3). After the 1960s, there was a distinct shift with spruce growth negatively correlated with maximum and minimum summer temperatures. This suggests growth was limited by the high summer temperatures in the 1960s, either directly by making it too warm for the trees to grow optimally; or indirectly because there was a concurrent large-scale drought during this decade. These trends are similar to *Cook, Johnson & Blasing (1987)* and *Mclaughlin et al. (1987)*, who observed that late summer temperatures (previous July maximum) negatively influenced red spruce radial growth in Northeastern forests.

### Precipitation

Prior to the 1960s, red spruce at all sites showed a positive response to precipitation and after the 1960s drought, MG and OM showed an increasingly negative correlation with

precipitation, while SF and MF continue to have a positive correlation with precipitation. The threshold response model proposed by *Cook & Johnson (1989)* is supported by these data. The 1960s drought represented a prominent climate event, after which precipitation was no longer a limiting factor at MG and OM sites, and increased precipitation was correlated with decreased ring-widths at these sites. High-elevation forests tend to be more limited by temperature (*Schweingruber, 1996*), and low-elevation sites are typically more sensitive to precipitation (*Cook & Johnson, 1989*). *Webb et al. (1993)* found that range-edge red spruce in bog sites were slower-growing but had more robust populations, while upland sites supported increasingly drought sensitive populations. In contrast, my study sites displayed decreasing drought sensitivity after the extensive drought of 1960. Only one forest, MF, showed a significant precipitation response in DendroClim, unique in comparison to the three other forests sampled.

Prior to 1960, growth at MF was positively correlated with precipitation from March of the previous year. After 1960, MF had a strong positive correlation with precipitation from the previous September and August of the current growing season with radial growth in red spruce at MF. The precipitation-growth response at MF shifts after the 1960s, similar to the pattern of temperature response which shifts in the 1960s, suggesting a precipitation threshold exists at MF. *Büntgen et al. (2006)* found a similar lack of stability between the growth/climate relationships for populations of Norway spruce, and suggest an increasing sensitivity to drought could be altering the climate-growth response due to increased evapotranspiration. While I did not directly measure rates of evapotranspiration, drought stress is a plausible explanation for declining radial growth of red spruce in some sites in Western Massachusetts forests-specifically induced by the 1960s drought, while other sites started to decline in the 1940s.

## Non-climatic environmental stressors

A variety of factors influence spruce growth in western Massachusetts where climate change is likely to have a large influence on populations at their range margin. The climatic-threshold theory states that trees shift their growth patterns and carbon allocation (i.e., ring widths) in response to climate conditions. This theory specifically predicates that climate is the main stressor that has induced physiological changes in tree growth (*Cook & Johnson, 1989*), and accurately describes the patterns in ring-widths observed at my sites although it is possible that forest dynamics are also contributing factors. In contrast, I would expect uniform responses to environmental triggers like pollution or acid deposition across all sites, which I do not see in Western Massachusetts. Pollution would also not discriminate across stand types (second-growth vs. old-growth) and some studies have shown the widespread decline to be especially prevalent in second-growth stands (*Van Deusen, 1987*).

Acid deposition has been shown to influence other red spruce populations, however, at these sites, I observed no visible symptoms of acid deposition (needle reddening or

winter injury) and it was not recorded historically in CFI notes. Additional foliar and soil chemistry studies could provide insights into nutrient cycling and other aspects of these red spruce forests, and should be pursued to tease apart the potential effects in Western Massachusetts. In New York and Vermont, acid deposition was considered to be more of a driving force for decline of red spruce, especially because of altered foliar and soil chemistry (*Battles et al., 2003*; *Mclaughlin et al., 1987*). In those high elevation forests, acid deposition may have exacerbated red spruce decline, and with the introduction of the Clean Air Act in 1970 a recovery of high-elevation red spruce populations would have been likely (*Van Deusen, 1990*). While this mechanism may be an important driver in other areas, we do not see evidence for acid deposition leading to decline in western Massachusetts red spruce populations. I observed that red spruce importance values were consistent over time, and found no evidence of a decadal-scale decline in basal area increment from the 1960's to the present, which would have supported an acid deposition explanation.

## CONCLUSIONS

Red spruce may be especially susceptible to climate change in Massachusetts because populations are isolated geographically, and may not have the topographic landscape conducive to tracking climate, which could lead to a range contraction (*Morin, Viner & Chuine, 2008*). Conversely, species such as tulip poplar (*Liriodendron tulipifera*) or pin oaks (*Quercus palustris*) which are at their northern range limit in Massachusetts may enjoy enhanced growth as climate continues to warm. Western Massachusetts red spruce chronologies demonstrate a changing climate-growth relationship. Temperature stress increases susceptibility to drought-induced growth reductions, making precipitation a limiting factor when temperatures increase. This shift from positive to negative growth correlations reinforces the hypothesis that the climate where these trees established has changed over the past 100 years, and is now warmer than ideal for growing conditions of red spruce. It is likely that a variety of co-factors have influenced red spruce populations, including a reduction of pollution levels associated with acid deposition, which might alleviate growth stress, and competition-induced red spruce mortality. Radial growth declines and reduced seedling and sapling abundance, suggests that red spruce may not maintain viable populations in Massachusetts forests at low-elevations. However, red spruce will persist at high-elevation sites like Mount Greylock, where spruce regeneration and basal area are increasing.

## ACKNOWLEDGEMENTS

Thank you to advisers Matthew Kelty, David Orwig, Neil Pederson, and Glenn Motzkin for assistance with design and analysis. Thank you to Israel Del Toro for field and laboratory assistance. Thank you to Henri Grissino-Mayer, Peter White, and Israel Del Toro for comments on earlier versions of this manuscript. Thank you to Kerry Woods, Daniel Gavin, and an anonymous reviewer for helpful revisions on this manuscript.

### Funding
Funding for this project was provided by a McIntire-Stennis Research Grant. The funders had no role in study design, data collection and analysis, decision to publish, or preparation of the manuscript.

### Grant Disclosures
The following grant information was disclosed by the author:
McIntire-Stennis Research Grant.

### Competing Interests
The author declares there are no competing interests.

### Author Contributions
- Relena Rose Ribbons conceived and designed the experiments, performed the experiments, analyzed the data, contributed reagents/materials/analysis tools, wrote the paper, prepared figures and/or tables, reviewed drafts of the paper.

### Field Study Permissions
The following information was supplied relating to ethical approvals (i.e., approving body and any reference numbers):

Massachusetts Department of Conservation and Recreation Special Use Permit.

### Supplemental Information
Supplemental information for this article can be found online at http://dx.doi.org/10.7717/peerj.293.

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
