# Peer review of "Disturbance and climatic effects on red spruce community dynamics at its southern continuous range margin"

_PeerJ, doi:10.7717/peerj.293_

## Round 0.1 · original submission · Major Revisions

I agree with reviewers that the combination of long-term plot data with dendrochronological techniques has great potential for understanding the complicated red spruce story. Consequently, I would hope to see a presentation of your data-set in a publishable form. However, several aspects of the current MS need careful and thorough attention before that's the case. I will not reiterate details of reviewers' comments (which are particularly thorough and thoughtful), but, in general, note several general themes that should guide your efforts in revision:

1) there are many instances of sloppiness and simple errors in prose and presentation. Some interfere with clarity and interpretation (e.g., apparent switching of Figs 2a and b?), but careful editing for clarity and grammatical correctness is called for throughout.
2) There are problematic oversights in placing this work in context of past and recent research. Most importantly, perhaps, the lack of any reference to massive body of research on effects of acid deposition is quite odd.
3) There are some important methodological questions and suggestions from reviewers regarding analysis, presentation, and interpretation of tree-core data.
4) (probably most importantly) there's a tendency towards over-interpretation of results to assert recent, temperature-related deterioration of performance of red spruce in these stands (e.g., trends shown in Fig. 6 being described as showing general 'trends of declining BAI' seems a stretch!).

A few specific points in addition to those from reviewers
* Table 2 gives only synthetic, relative IV; actual values for BOTH basal area and density would be much better. The combined and relativized IV's are not very helpful -- they throw away a lot of information for no good reason.
* Why only 3 sites in Fig. 2?
* line 319: "too warm..." But you earlier noted drought over this period as likely driver?
* lines 203-204: "negative relationship with precipitation in October", but "no significant responses to precipitation" seems contradictory?

Reviewer 1 ·

Basic reporting

I noted some errors in figures, and thus could not examine the plot data. Several paragraphs could use some re-organization, as the topic jumps around too much. The Results/Disucssion could benefit from an an overall re-outline, resulting in a moderate amount of work. I think the readers would benefit from making very distinct the sections related to growth trends, disturbance, and how this might relate to the plot data, then the details of the growth-climate relationship and the shifting relationships, then finally make a coherent argument about stand dynamics especially regarding how much growth declines (as presented in Figure 8) might be a result of climate stressors versus stand dynamics (i.e., canopy closure and competition).

Experimental design

The sampling and scope of the study is excellent and represents a very large project to be undertaken by one person.

Validity of the findings

The findings appear to be valid based on my readings--with the exception of site MF where there might be poor cross-dating.

Additional comments

This is a very important study that bridges plot resurvey data with tree-ring reconstructions of disturbance and climate influence on tree-growth. It represents a very large amount of work and addresses a complex topic that has been batted about for 25 years. Thus, there is a plethora of precedent and previous attempts to interpret red spruce tree-ring data. However, the there is certainly a need to use long-term plot data alongside tree-ring data to help explain growth trends observed in the tree-ring data. This is a well-designed study that takes this approach. My comments below are a mix of specific comments and broad suggestions for reorganization.

“We” is used in two places but there is one author.

Lines 39-45: This distillation of the causes of red spruce decline makes no mention of acid precipitationm which would increase the sensitivity to winter injury. There could then be a rationale for a recovery of populations if there is less pollution today.

Line 52-53 needs citations. I know some studies where this was not found.

Line 109: The J2X software, or the Velmex system, does not connect to a microscope.

Figure 2: The values in this graph seem strange. Red spruce basal area is greater than total basal area at all sites. If 2a and 2b were swapped, it still does not make sense as the values in 2b + 2c should equal 2a. Fig 2d and 2e are identical. Also, I prefer the convention that the y-axis begins at 0.

MF is not on Fig 2, but the results are discussed in the text.

Because of the seeming errors in Fig 2, I can’t interpret the text results. For example, this sentence: “Non-spruce basal area declined by at least 50% at MG and MF forests, increased slightly at SF and increased by over 100% at OM. “ does not correspond to the figure. I will not comment further on the stand composition results at this point.

Line 172: “Mean maximum, minimum, and summer temperatures”. “Summer” should not be part of the list…I assume you meant “mean maximum and mean minimum summer temperature”.

Line 176-177: This sentence seems out of place—it is a single-sentence summary of a lot of results, while the emphasis before and after is on the gross patterns of the growth and climate history.

Lines 179: “While tree growth naturally slows after maturity, the removal of age-related growth trends through detrending clarified radial growth trends unrelated to age.” Detrending removes all growth trends over a certain frequency (32 years in your case), not just those related to age. Removing only the age-related trend at the stand scale would require the RCS method. So detrending is not clarifying any long-term trend; it is emphasizing the decadal-to-annual variability in growth rates.

Line 185: The use of BAI was not introduced in the methods.

In general, the first two paragraphs of “Radial growth and climate response” could be reorganized to address the results with increasing levels of detail (and detrending). It appears the author wished to first just mention all the results briefly before delving into longer explanations and interpretations, but I found it confusing. I suggest to remove these paragraphs and begin with a section on “Forest-level trends in BAI and disturbance history”—this section would describe the stand-level histories. Then the next major section would address the growth-climate correlation (the Dendroclime and RFA results). (It is not clear by BAI and RFA were discussed in the same section as they are very different analyses). The justification for detrending should be in the methods, not here. Separation of results and discussion may aid in organization. However, the above strategy may still work with combined results/discussion.

Figure 3-6: The green background is confusing because it could be interpreted as ‘0’ values on the color scale. However, Dendroclime only reports significant results, so the green is actually “non-significant” correlations.

Figure 6: MF shows very little interannual variability, suggesting poor cross dating. I did not see the supplementary material so did not see the raw data. This is probably why there are poor correlations emerging (lines 235-237).

309-310. The classic cite for the unstable growth response of red spruce is Johnson et al. 1988 in PNAS.

The finding of growth declines in the 80s and 90s and distinct releases in the 20s and 30s is a good finding of this study. Such findings support the Reams and Vandusen studies that contend spruce decline is a product of forest dynamics: earlier growth releases must eventually slow down, and competition results in some mortality.

The concluding sentences on lines 368-370 are interesting. It could be driven home that the major evidence for this is 1) maintainence of importance values of red spruce through time, and 2) lack of a decadal-scale decline in BAI from the 60s to present.

The other robust finding at the end of the conclusions is that the reversal from positive to negative growth-temperature correlations suggests a change from temperature-limiting to temperature-stressing conditions. It is good other tree-ring studies have supported this (cites Cook et al. 1987 and McLaughlin et al 1987) but Gavin et al. did not find this, but showed consistent correlation with GDD through most of the record (though using different methods). A physiological study supporting the summer tmax stressor would be helpful, if one exists.

Reviewer 2 ·

Basic reporting

I believet hat there were several problems with the context and coceptual framework behing this study. These included:

1) The context: the author indicates that the range-wide decline (reduced growth and increased mortality) of red spruce that stared in the 1960s was cause by “climate change and a combination of environmental disturbances”. However, hundreds of peer-reviewed articles and several books all conclude that red spruce decline was instigated by inputs of acid deposition. Indeed, red spruce decline is the best documented example of the connection between acid deposition and tree health/productivity declines for North America. The omission of this basic context is surprising and it sets up a basic failing of the study – the establishment of flawed or incomplete conceptual frameworks and working hypotheses for the study.

2) Conceptual frameworks: given the broad literature base on red spruce’s sensitivity to acid deposition, the most logical primary conceptual framework for assessing differences in red spruce growth would be to assess patterns relative to the extent of acid loading (temporally and spatially). This should have been one of the hypotheses tested. Because red spruce decline was also associated with foliar winter freezing injury (which is exacerbated by exposure to acid deposition), a logical secondary hypothesis would be that climate (e.g., changing exposure to lethal freezing temperatures during winter) may also help account for differences in growth over time. Because there is some evidence that growing season weather patterns may also predispose red spruce to winter injury the next winter (e.g., Schaberg et al. 2011), then assessing influences of weather parameters at other times of the year would also be justified (perhaps a third hypothesis). These sorts of hypotheses have far greater grounding in the large literature base pertinent to red spruce decline than the ones currently stated. I also question the conceptual framework that existing red spruce populations represent the southernmost continuous range of the species (and are therefore an ideal test population for assessing the impacts of climatic warming). Based on the widely referenced rage maps presented by Little (1972), the contiguous range of the species dips farther south in NY (by perhaps 150-200 miles?) than it does in MA. More importantly, there is no reason to believe that the discontinuous nature of the species range below NY represents a response to climate. Instead, I suspect that this reflects the broad scale clearing of the species and land conversion away from forests that persisted through the 19th and 20th centuries.

Experimental design

There were several problems with the studies design and methods. These included: 1) Sample size – 550 cores were collected, but only 412 cores were used for disturbance analysis and only 213 were used for climate analyses. The author suggests that “core defects” and problems with cross-dating caused this reduction in sample size. However, this high attrition rate is quite unusual and suggests that some serious problems in dendrochronological methods must have occurred.

2) Especially when assessing connections between climate and tree ring growth, a statistic called the expressed population signal (EPS) is calculated for annual growth data. The EPS is a measure of the common variability in a chronology and is dependent on sample size: when it falls below a predetermined value, typically 0.85, the chronology is a less reliable indicator of a coherent stand-level signal (Wigley et al. 1984). This provides a statistical threshold of when (at what date) a growth chronology should begin. In contrast, the author truncated chronologies “when fewer that two core replicates” existed (see Figure 6). It is highly questionable that two cores cold provide an adequate representation of stand growth dynamics.

3) DendroClim data (Figures 3, 4 and 5) are only semi-quantitative. They do portray the results of quantitative correlation analyses. However, the color-based representation of results then must be interpreted qualitatively to evaluate patterns of correlation results (consistently positive or negative correlations) over time. In contrast, stepwise linear regressions probably would have provided a better, strictly quantitative analysis of associations between growth and climate parameters (and would have been preferable).

Validity of the findings

There seemed to be a conflict between the paper’s conclusions and the existing literature/new data presented: the paper concludes that “there has been a negative growth response to regional warming, as spruce radial growth was mostly constrained by increasing temperatures”. However, recent papers (not referenced in this paper) show that red spruce woody growth (Kosiba et al. 2013), biomass and regeneration (van Doorn et al. 2011) is now on the upsurge. Indeed, recent woody growth from 2007 through 2010 is almost two-times the average for growth over for these same trees over the last 100 years (Kosiba et al. 2013). Indeed, the data evaluated by Kosiba et al. (2013) included growth data from two plots on Mount Greylock – one of the four sites evaluated for the paper under review. Even the new data presented in this paper (Figure 6) seems to show greater general growth now than earlier time periods. Isn’t the general pattern of greater growth (van Doorn et al. 2011, Kosiba et al. 2013, Figure 6) as temperatures have tended to increase inconsistent with the notion that there has been a negative growth response to regional warming?

References cited here and previopusly:

Kosiba, A.M., P.G. Schaberg, G.J. Hawley, C.F. Hansen. 2013. Quantifying the legacy of foliar winter injury on woody aboveground carbon sequestration of red spruce trees. Forest Ecology and Management 302:363-371.

Little, E.L. 1971. Atlas of united states Trees. Volume 1. Conifers and important hardwoods. U.S. Department of Agriculture, Forest service, Miscellaneous Publication No. 1146.

Schaberg, P.G., B.E. Lazarus, G.J. Hawley, J.M. Halman, C.E. Borer, C.F. Hansen. 2011. Examination of weather-associated causes of red spruce winter injury and consequences to aboveground carbon sequestration. Canadian Journal of Forest Research. 41:359-369.

van Doorn, N.S., J.J. Battles, T.J. Fahey, T.G. Siccama, P.A. Schwartz. 2011. Links between biomass and tree demography in a northern hardwood forest: a decade of stability and change in Hubbard Brook Valley, New Hampshire. Canadian Journal of Forest Research. 41:1369-1379.

Wigley, T.M.L., K.R. Briffa, P.D. Jones. 1984. On the average value of correlated timeseries, with applications in dendroclimatology and hydrometerology. Journal of Climate and Applied Meteorology 23:201-213.

---

## Round 0.2 · Minor Revisions

I appreciate your efforts in response to the first round of reviews. The same reviewers indicate that you have addressed at least some of their concerns effectively. However, both raise some questions and make some useful suggestions for the current manuscript.

I would like to see this study published in PeerJ, and believe the current manuscript is nearly ready for acceptance, but I believe that one more round of revision will bring substantial benefits. Some of the reviewers' specific suggestions seem straightforward to me; others will be a matter of judgment. Overall, I believe the paper could still benefit from work towards reduction in length, some of which could be accomplished by reducing redundancies between sections.

I hope this revision will result in rapid acceptance

Reviewer 1 ·

Basic reporting

See General Comments for the Author (below).

Experimental design

See General Comments for the Author (below).

Validity of the findings

See General Comments for the Author (below).

Additional comments

I reviewed an earlier version of this manuscript and, although changes have been comparatively minor, I feel the current re-write is an improvement overall. For example, the paper now more directly addresses the influence of acid deposition on red spruce health and productivity. That said, there are still multiple issues that I believe should be addressed before the paper would be ready for publication. These issues are described below in association to specific line numbers and Figures /Tables related to the areas to be addressed.

Line 13 and again in line 330: red spruce can be a component of the boreal forest, but it is not a boreal species – it is a temperate conifer (Strimbeck et al. 2007).

Lines 23-25: How does this relate to red spruce in MA?

Lines 30-35: Here and elsewhere the author now does a better job of acknowledging the role of acid deposition in influencing the growth and health of red spruce. However, there is still an underrepresentation of the influence of acid deposition relative to other factors – like climate. For example, see the Springer-Verlag Ecological Studies Book 96 – Ecology and Decline of Red Spruce in the Eastern United States (reference below). It would be more in keeping with the science on red spruce decline to acknowledge that acid deposition was a significant contributor to changes in red spruce productivity during the latter half of the last century, but that the emphasis on this one stressor has limited analysis of other factors (like changing climate) that may be of greater influence as pollution leading decreases following implementation of the 1990 Amendments to the Clean Air Act?

Lines 3-45: Associated with my last point, it seems odd that the two hypotheses are 1) disturbance including acid deposition and many other factors, and 2) climate change. Perhaps it would be better to acknowledge that scientific focus has been on the influence of acid deposition, but that the influence of disturbance and climate change have received far less specific analysis (especially recently when climate change is accelerating) – thus they are the emphasis in this study?

Lines 113-120: The author should describe how it was objectively determined the date at which increment core chronologies were considered robust enough to be valid to be included in climate and other comparisons? One method is to calculate the Expressed Population Signal (EPS). The EPS is a measure of the common variability in a chronology and is dependent on sample size: when it falls below a predetermined value, typically 0.85, the chronology is a less reliable indicator of a coherent stand-level signal (Wigley et al. 1984). This provides a statistical threshold of when (at what date) a growth chronology should begin. This will help establish if the x-axes in Figures 3, 4, 5 and 6 are appropriate or not (e.g., I doubt that the sample sizes for growth lines before 1930 or so in Figure 6 have a sufficient sample size to be truly representable of the stands overall). Also, in Figures 3 and 4 the x-axis varies among the stands – this is problematic when comparing trends over time.

Lines 152-163: It is unclear to me how one related declines lasting 10- 15 years to the number of damage events per decade – the time scale seems to be at odds?

In general, there are many redundancies among the sections on “Radial growth and climate responses”, “temporal stability in climate-growth responses at a regional and forest scale”, “Non-stable climate growth relationships” and “Precipitation”.

Lines 370-380: There are expected to be site-to-site differences in the influence of acid deposition on tree health and productivity. This is due to variation of deposition with longitude and elevation (more acid deposition toward the west and at higher elevations) combined with site-specific differences in soil nutrition (especially base cations). That is why researchers increasingly use spatially explicit models of critical loads (a nutrition index) and exceedance (a pollution index) to project how these vary across the landscape (e.g., Duarte et al. 2013).
Lines 381-383: These are likely incomplete indicators of possible damage at these sites. For example, the analysis of foliar winter injury and loss (Lazarus et al. 2004) and associated reductions in woody growth (Kosiba et al. 2013) following the region-wide winter injury event in 2003 included two elevations at Mount Greylock, MA ( a location used in this study. Foliar winter injury and loss and subsequent reductions in woody growth were both substantial at the Mount Greylock plots.

Table 1: the unequal sampling size (from 1 to 7 plots per location) means that mean and variance estimates will have different levels of precision among the locations. This may have contributed to the differential relationships among growth and climate parameters that are now attributed to “location” but which may, in part, be an artifact of differences in sample sizes?

Tables 2 and 3: Shouldn’t there be some listing of statistical results in these tables so the reader can evaluate if the means are different from each other? I assume that these are means (and standard errors in parentheses in Table 3)? Here and elsewhere, Table and Figure captions could be more comprehensively descriptive.

Figure 2: are these means (circles) and standard errors (vertical lines)?

Figures 3, 4 and 5: In the captions, better define what the upper versus lower case representations of month abbreviations stand for.

Figure 6: As per my earlier comment, some better means of assuring that there is an adequate sample size so that growth trends are reliable (and not based on just two cores) needs to be employed here. I truly doubt that the early growth data from any of these sites has a large enough sample size to be ecologically relevant and robust. Please consider using the EPS method for deciding the “start date” for a stands chronology.

Figure 7: Indicate what the months shown in lower case letter represent.

Figure 8: All the axes on this figure have issues – the left y axis is labeled oddly (can you have a negative % tree disturbance event?), the x axis should not include dates with very low sample size (too few cores make for poor estimates), and the right Y axis has no scale or label at all.

References cited:
Duarte, N., L.H. Pardo, M.J. Robin-Abbott. 2013. Susceptibility of forests in the northeastern USA to nitrogen and sulfur deposition: critical load exceedance and forest health.

Eagar, C., M.B. Adams (Eds) 1992. Ecology and decline of red spruce in the eastern United States. Springer-Verlag, New York, 417p.

Kosiba, A.M., P.G. Schaberg, G.J. Hawley, C.F. Hansen. 2013. Quantifying the legacy of foliar winter injury on woody aboveground carbon sequestration of red spruce trees. Forest Ecology and Management 302:363-371.

Strimbeck, G.R., T.D. Kjellsen, P.G. Schaberg, P.M. Murakami. 2007. Cold in the common garden: comparative low-temperature tolerance of boreal and temperate conifer foliage.

Wigley, T.M.L., K.R. Briffa, P.D. Jones. 1984. On the average value of correlated timeseries, with applications in dendroclimatology and hydrometerology. Journal of Climate and Applied Meteorology 23:201-213.

Reviewer 2 ·

Basic reporting

No Comments

Experimental design

No Comments

Validity of the findings

No Comments

Additional comments

Abstract: Wording suggestions:
This study uses 50 years of population data….
However, among the 17 plots, there was no consistent trend in basal area or
density for red spruce or other woody species.

Fig. 2 is mostly fixed….but now 2b and 2c are identical plots. I assume Table 2 is correct.

Seems strange sentence on Fig 2d is at end of paragraph, and sentence on 2e and 2f strts the next paragraph. Why not keep all discussion of density in one paragraph (strive for parallel structure).

Why is the sentence on BAI on lines 198-200, then followed by RFA analysis (a completely different analysis), and then the topic is back on BAI on line 206? Why is climate correlation (RFA) merged with BAI trends in this section?

LINE 295-296: “Climate or physiology” is a false dichotomy….climate affects tree physiology. Be more specific.

In the conclusions section—one more caveat to add is that the trees we observe today are the ones that survived the 1980’s. We can’t reconstruct growth declines of the trees that have died. So, changing perspective on the decline from a study made today and a study made in the 1980s must be sensitive to the fact the populations have changed and there has been selection for the hardier trees (note decline in spruce density).

---

## Round 0.3 · accepted · Accept

Thanks for your work in revising this manuscript